# Effect of Annealing Ambient on SnO$_2$ Thin Film Transistors Fabricated via An Ethanol-based Sol-gel Route

**Hyunjae Lee [1], Seunghyun Ha [1] , Jin-Hyuk Bae [1] , In-Man Kang [1], Kwangeun Kim [2] , Won-Yong Lee [1],\* and Jaewon Jang [1],\***

[1]   School of Electronics Engineering, Kyungpook National University, Daegu 41566, Korea
[2]   Department of Electronic and Electrical Convergence Engineering, Hongik University, Sejong 30016, Korea
\*   Correspondence: yongsz@knu.ac.kr (W.-Y.L.); j1jang@knu.ac.kr (J.J.)

**Abstract:** The effect of annealing ambient on SnO$_2$ thin-film transistors (TFTs) fabricated via an ethanol-based sol-gel route was investigated. The annealing ambient has a significant effect on the structural characteristics and chemical composition and, in turn, the device performance. Although the crystalline-grain size of the SnO$_2$ films annealed in air was the smallest, this size yielded the highest field-effect mobility. Compared with the minimization of boundary scattering via crystalline-size increase, augmentation of the free carrier concentration played a more critical role in the realization of high-performance devices. The fabricated SnO$_2$ TFTs delivered a field-effect mobility, subthreshold swing, and on/off current ratio of 10.87 cm$^2$/Vs, 0.87 V/decade, and 10$^7$, respectively.

**Keywords:** Sol-gel; SnO$_2$; thin film transistor; annealing ambient

## 1. Introduction

Oxide semiconductor-based thin-film transistors (TFTs) are characterized by high electron mobility, a wide band gap, and applicability to various high-performance and transparent devices. The demand for these TFTs has increased. Applications requiring TFTs include see-through displays, transparent sensors, and electrochromic windows [1–4]. Oxide semiconductor-based TFTs are expected to supersede the conventional Si-based TFTs. Therefore, oxide TFTs based on Indium-based metal oxides have been extensively studied in the last decade. However, indium is a costly and rare metal that is found in only a few mining spots worldwide [5] and, hence, a new metal oxide system (without rare metals such as indium) is required. Representative n-type metal oxide semiconductors include ZnO, In$_2$O$_3$, and SnO$_2$. Among these materials, SnO$_2$ has the largest band gap, and is characterized by a high intrinsic mobility and low melting point. These features are attractive for applications that require high performance and transparent materials [6–8]. Various deposition techniques can be used to fabricate metal oxide films [9–11]. Unfortunately, these fabrication techniques require complex equipment and result in high manufacturing costs. Alternative techniques, such as solution processing, are therefore necessary. In particular, the solution processing-based fabrication of oxide TFTs is simple, low cost, and compatible with spin coating and inkjet printing [12–14]. A sol-gel process, i.e., one of the representative solution processes, is simple, easily controlled, and represents a cost-effective method of forming metal-oxide films. The quality of the final-formed film can be strongly influenced by precursors, substrate types, annealing temperature, and annealing conditions.

The present investigation was aimed at elucidating the effect of annealing ambient on the electrical performance of SnO$_2$ TFTs fabricated by means of a sol-gel method. Annealing conditions play a critical role in determining the defect sites formed inside the material and, hence, have a significant influence on the device performance [15–17].

## 2. Materials and Methods

Tin (II) chloride dihydrate ($SnCl_2 \cdot 2H_2O$, 0.001 mol) purchased from Sigma Aldrich (St. Louis, MO, USA) and was used as the precursor. The $SnCl_2$ precursor was dissolved in 20 mL ethanol, thereby yielding a 0.050 M solution. $SnO_2$ thin films were obtained by spin-coating (3000 rpm, 50 s) the Si/SiO₂ substrate, which was previously cleaned during a 60-min ultraviolet/ozone (UV/$O_3$) treatment. The fabricated films were dried in air for 10 min on a hot plate at 150 °C. Afterwards, the annealing process (heating for 4 h at 600 °C in a tube furnace) was executed under three different ambient conditions (air, vacuum, $N_2$). The $N_2$ gas flow into the furnace was maintained at a rate of 2.5 L/min, and the pressure inside the furnace was kept below 0.1 MPa under vacuum conditions. The $SnO_2$ TFTs were fabricated on thermally grown 100-nm-thick $SiO_2$/Si substrates. Moreover, 50-nm-thick Au electrodes were deposited as the bottom source and drain electrodes (channel width: 1000 μm, channel length: 100 μm) by using an e-beam evaporator and a lift-off process. The phase and structural characteristics of the films were determined via Grazing Incidence X-ray Diffraction (GIXRD, incident angle: 0.3°) with CuKα radiation (wavelength: 1.54 Å). The elemental composition and chemical state of the annealed films were evaluated by means of X-ray photoelectron spectroscopy (XPS; Quantera SXM (Physical Electronics, Chanhassen, MN, USA), chamber pressure $4 \times 10^{-7}$ Pa). The optical properties were obtained at wavelengths ranging from 350 nm to 950 nm, using a LAMBDA 265 UV/Vis spectrophotometer. The surface morphology of the films was examined via scanning probe microscopy (SPM; Park NX20 (Park Systems, Suwon, Korea), tapping mode). In addition, the electrical properties of the TFTs were measured in air using a Keithley 2636B semiconductor parameter(Keithley Instruments, Cleveland, OH, USA) analyzer and a probe station.

## 3. Results and Discussion

The prepared thin films were annealed at 600 °C for 4 h under three different ambient conditions: air, vacuum, and $N_2$. The GIXRD spectra of the crystallized films, Figure 1a, correspond to the tetragonal phase of $SnO_2$ (JCPDS 41-1445), i.e., polycrystalline tetragonal $SnO_2$ occurred in each annealed film. Diffraction peaks at 26.6°, 33.8°, 37.9°, and 51.7° correspond to the (110), (101), (200), and (211) crystal planes, respectively, of this phase. The strongest peak intensities were observed for the TFT annealed in $N_2$. The full width at half-maximum (FWHM) of a given GIXRD peak is closely correlated with the crystallite size of a particular crystal orientation. The FWHM of the $SnO_2$ (110) peak was narrower than those of the other peaks, suggesting that $SnO_2$ crystallites grew initially in the (110) plane.

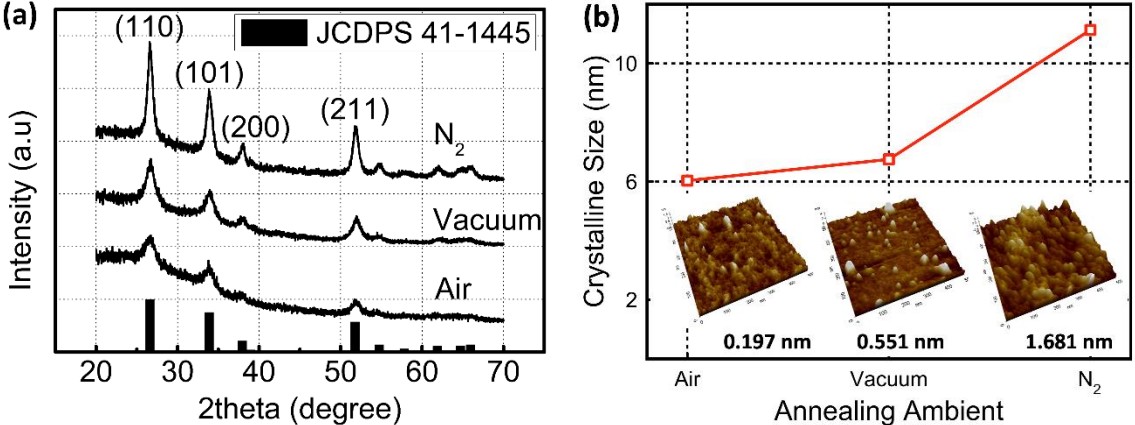

**Figure 1.** (**a**) GIXRD patterns and (**b**) calculated crystalline sizes of the $SnO_2$ films annealed in air, vacuum, and $N_2$. The inset shows the SPM images.

The crystalline size of the films (see Figure 1b) was calculated from the Scherrer equation, which is given as follows:

$$D = \frac{0.9\lambda}{\beta \cos \theta} \tag{1}$$

where, D, $\lambda$, $\beta$, and $\theta$ denote the crystalline size, CuK$\alpha$ wavelength (1.54 Å), full width at half-maximum of the peak, and peak position, respectively. Measured $\beta$ values of 0.0236, 0.0211, and 0.0128 yielded D values of 6.03, 6.74, and 11.13 nm for the (110) phase of the SnO$_2$ thin films annealed in air, vacuum, and N$_2$, respectively. As the SPM images and root mean square (RMS) values of the films show (see inset of Figure 1b), the crystalline-grain size of the film annealed in N$_2$ is significantly larger than that of the films annealed in vacuum and air.

The chemical composition of films annealed under different ambient conditions was determined via by XPS. The Sn composition was determined from the Sn 3d$_{5/2}$ spectra (see Figure 2). The Sn 3d$_{5/2}$ spectrum consists of two peaks, which occur at 485.51 and 485.96 eV and correspond to Sn$^{2+}$ and Sn$^{4+}$, respectively. The O 1s peak is split into three peaks associated with SnO (Sn$^{2+}$), SnO$_2$ (Sn$^{4+}$), and hydroxide (OH groups) at 529.3, 530, and 530.9 eV, respectively. The Sn$^{4+}$/Sn$^{2+}$ ratio of the films annealed in air is considerably higher than those of the films annealed under other conditions. Compared with the amounts converted under other conditions, more SnO is converted into SnO$_2$ during annealing in air.

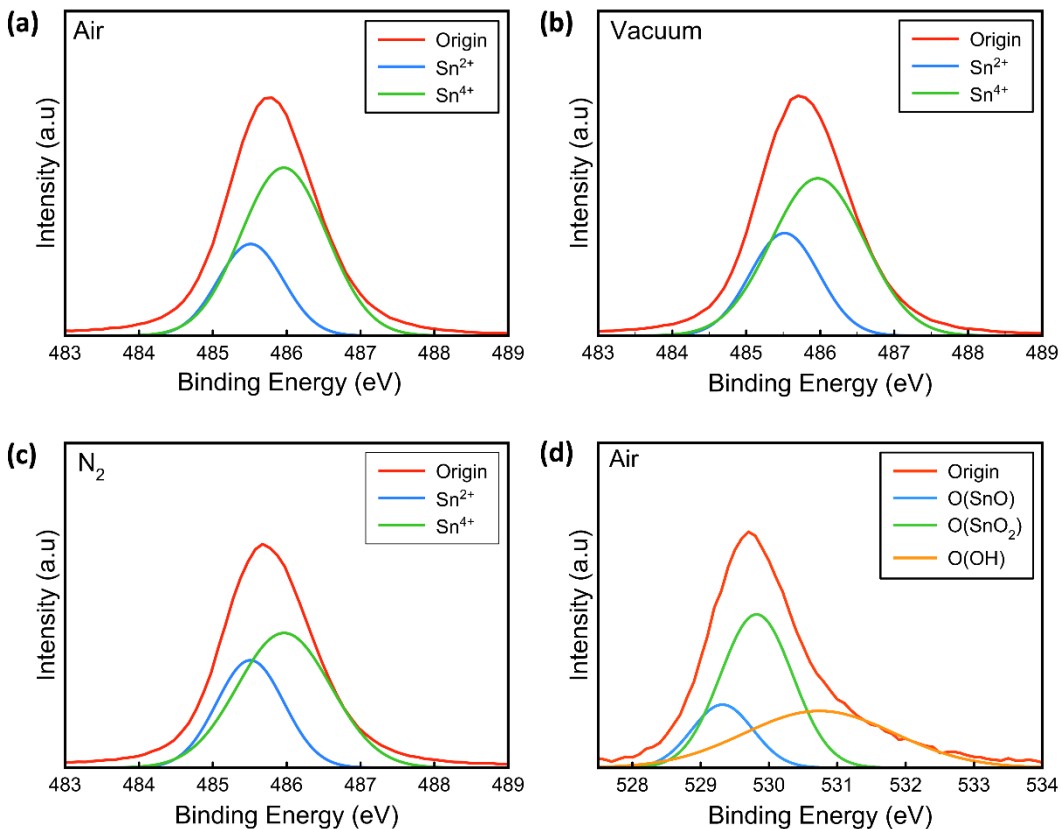

**Figure 2.** *Cont.*

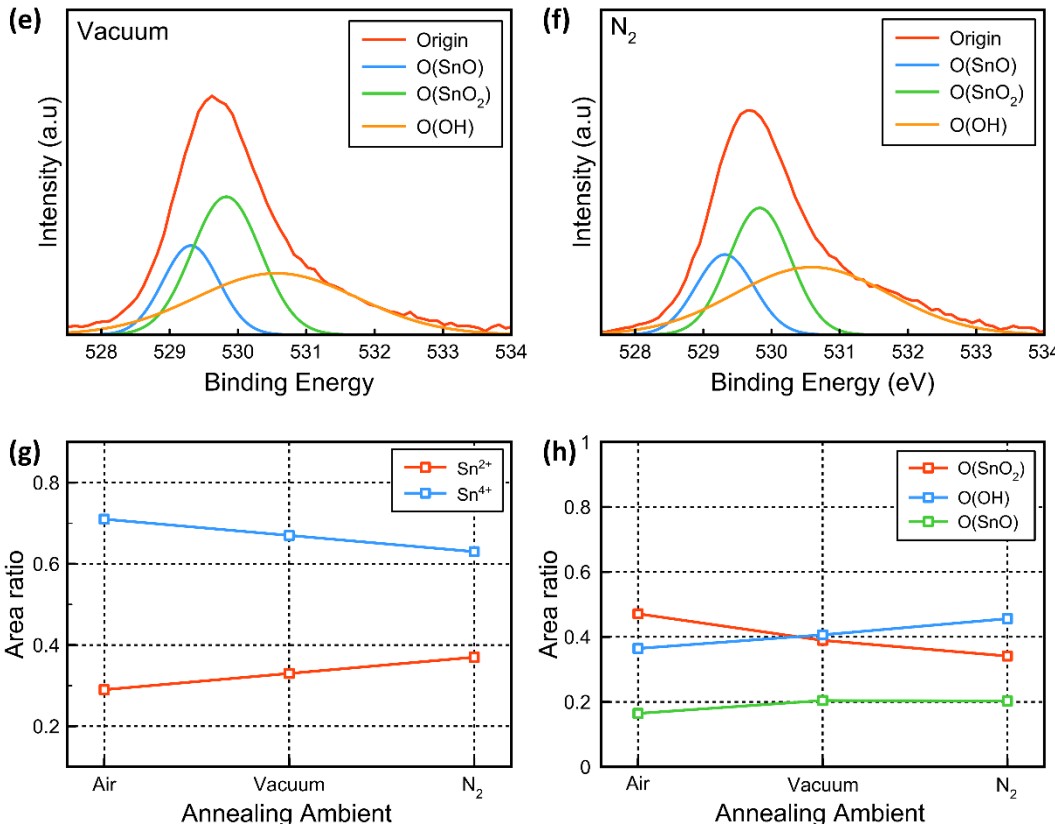

**Figure 2.** (**a**)–(**c**) Sn 3d$_{5/2}$ XPS spectra and (**d**)–(**f**) O1s XPS spectra of SnO$_2$ films annealed in air, vacuum, and N$_2$. Sn 3d$_{5/2}$ (**g**) and O 1s (**h**) compositions of films annealed in different annealing ambient conditions.

Figure 3 shows the electrical characteristics of thin-film transistors (TFTs) fabricated under different annealing ambient conditions (i.e., air, vacuum, and N$_2$). The fabricated devices have a bottom gate and bottom source/drain structure. The gate is p+ heavily doped Si and the gate dielectric is thermally grown SiO$_2$. Furthermore, the source/drain electrodes are composed of Au. Figure 3a–c shows the output (I$_{DS}$-V$_{DS}$) characteristics of the fabricated SnO$_2$ TFTs exposed to the different annealing ambient conditions. The I$_{DS}$-V$_{DS}$ curves are probed at gate voltages of −30, −20, −10, 0, 10, 20, and 30 V. In the low V$_{DS}$ region of the curves, I$_{DS}$ changes non-linearly with V$_{DS}$. This resulted from the relatively high work function of Au, compared with that of SnO$_2$, and the consequent Schottky contact between the SnO$_2$ channel and the Au source/drain electrodes [18–20]. The negative output differential resistance effect observed in Figure 3b was induced by the large gate leakage current. Figure 3d–f presents the transfer (I$_{DS}$-V$_{GS}$) characteristics of the fabricated SnO$_2$ TFTs exposed to different annealing ambient conditions and a biased drain voltage of +30.0 V and +1.0 V. The fabricated TFTs exhibited conventional n-type semiconductor properties with negative turn on voltage (V$_{on}$) and a normal on state (depletion mode). This indicates that excessive free carriers occur in sufficient numbers in the fabricated SnO$_2$ thin films, leading to easy formation of the channel at an applied gate voltage of +0.0 V. The performance of the fabricated SnO$_2$ TFTs is evaluated via four representative parameters, i.e., the field-effect mobility, on-current, subthreshold swing, and turn on voltage (V$_{on}$). The field-effect mobility in the saturation region can be calculated from the saturation current of the transistor:

$$I_{DS} = \mu C_i \frac{W}{2L}(V_G - V_{th})^2 \tag{2}$$

where, C$_i$, W, L, and V$_{th}$ denote the gate oxide capacitance per unit area, width of the channel, length of the channel, and threshold voltage, respectively. The calculated field-effect mobility and on-current

value are plotted in Figure 5. Mobility values of 10.87, 0.62, and 0.30 cm$^2$/Vs, and on-current values of $3.5 \times 10^{-3}$, $3.89 \times 10^{-5}$, and $5.44 \times 10^{-5}$ A were obtained for the TFTs annealed in air, vacuum, and N$_2$, respectively.

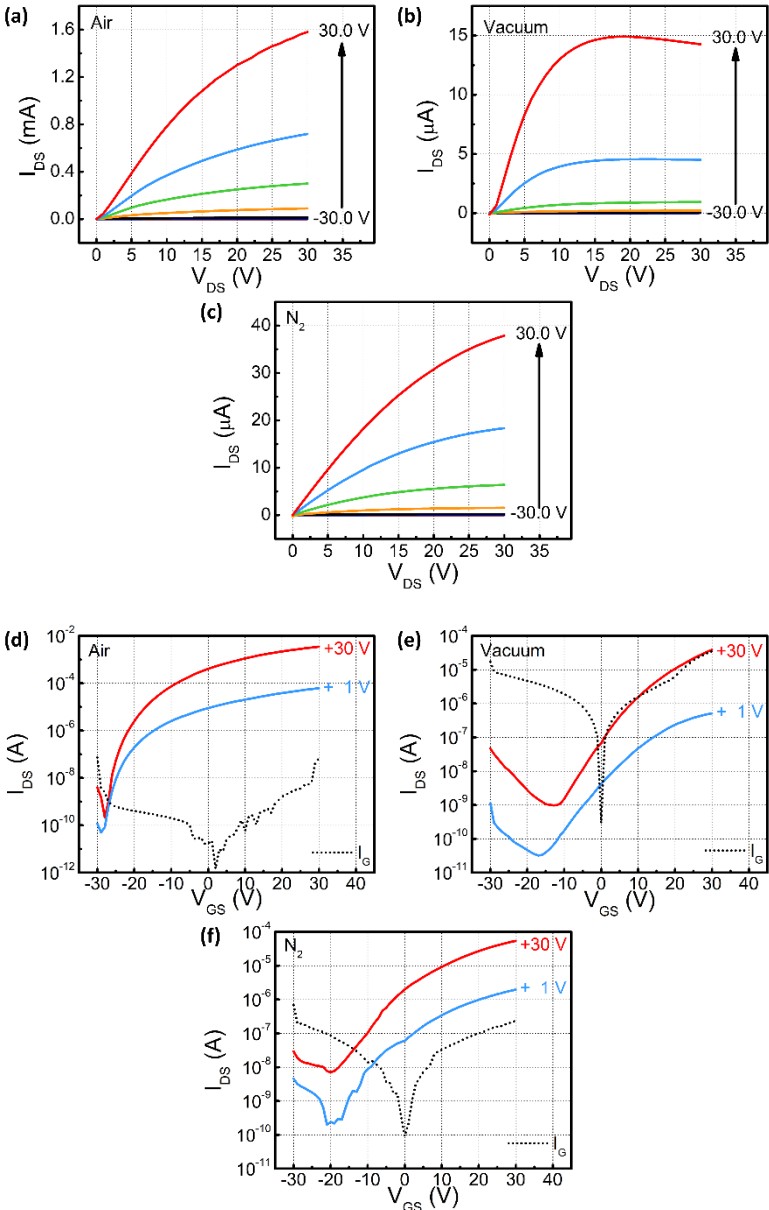

**Figure 3.** I$_D$-V$_D$ curves obtained for TFTs consisting of SnO$_2$ films annealed in (**a**) air, (**b**) vacuum, and (**c**) N$_2$. The corresponding transfer curves of SnO$_2$ TFT annealed in (**d**) air, (**e**) vacuum, and (**f**) N$_2$ with different V$_{DS}$.

The crystalline size of the active channel layer comprising TFTs plays a critical role in simultaneously improving the on-current and field-effect mobility via the suppression of boundary scattering [21]. This scattering decreases progressively with increasing crystalline size, thereby resulting in improved device performance. However, a different result was obtained in the present study. The smallest crystalline-grain size was calculated from the GIXRD data and SPM images of the SnO$_2$ films annealed in air, but these films exhibited the highest field-effect mobility in the saturation regime (10.87 cm$^2$/Vs). The effect of annealing condition on the device performance can be explained by determining the chemical composition of the SnO$_2$ films. Furthermore, XPS analysis of the films revealed that the

$Sn^{4+}/Sn^{2+}$ ratio of the films annealed in air is considerably larger than those of the films annealed under other conditions. Compared with that generated under other conditions, more SnO was converted into $SnO_2$ during annealing in air. It is well known that there are two binary stoichiometric tin oxides, SnO and $SnO_2$. SnO shows p-type semiconductor properties, and $SnO_2$ show n-type semiconductor properties. In this system, according to the XPS data, the largest $Sn^{4+}/Sn^{2+}$ ratio was obtained for the films annealed in air. This means that the decrease in hole carriers generated in the p-type SnO semiconductor compensate less due to the increase in electron carriers generated in the n-type $SnO_2$ semiconductor at the same time. The increase in the $Sn^{4+}/Sn^{2+}$ ratio indicated that the channel conductivity of the n-channel layer was enhanced and, consequently, the number of carriers increased. This results in strong n-type semiconductor properties with high electron carrier concentrations.

Figure 4 shows the energy level diagram determined from XPS and absorption spectra. The estimated bandgaps are 3.75, 3.80, and 4.17 eV, respectively. The conduction bandgap offset ($\Delta E_{CB} = E_g - \Delta E_{VB}$) of the $SnO_2$ films can be estimated from the XPS-measured valence band and absorption spectra. This value, which is strongly correlated with the carrier concentration [22], was lowest for the $SnO_2$ films annealed in air, indicative of the high carrier concentration characterizing these films. The major carrier transport mechanism operating inside the metal oxide semiconductors, i.e., percolation conduction, is improved by filling the trap state at high carrier concentrations [23]. The improved field-effect mobility and on-current originated from a significant increase in the free carrier concentration, and the calculated mobility values concurred with the XPS results. Therefore, compared with the crystalline size, the chemical composition played a greater role in determining the field-effect mobility in the saturation region and the on-current of the fabricated $SnO_2$ devices.

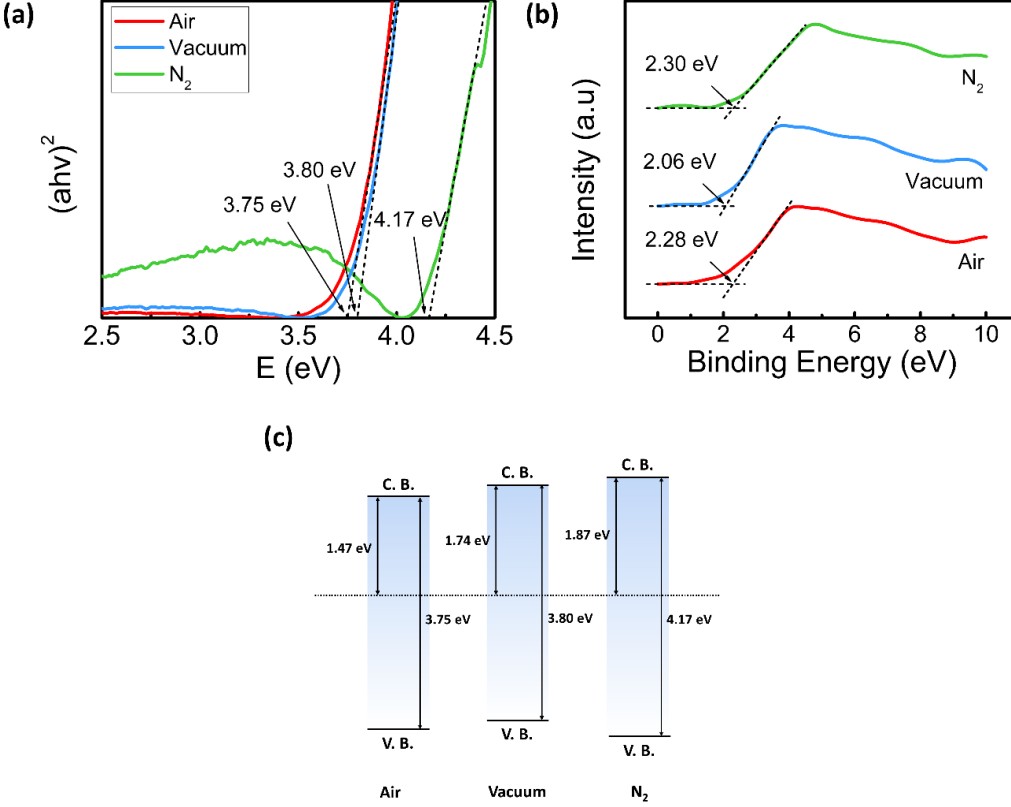

**Figure 4.** (**a**) Dependence of $(\alpha h v)^2$ on the photon energy (hv) of $SnO_2$ films annealed in air, vacuum, and $N_2$. (**b**) XPS spectra collected from regions near the valence band. (**c**) Energy level diagram including the relative energy position of the Fermi level (EF) with respect to the conduction band minimum and valence band maximum.

Moreover, the turn on voltage, and subthreshold swing (SS) were measured from the transfer characteristics and plotted in Figure 5b,c. Estimated voltage of −28.0, −11.0, −19.0 V, and SS values of 0.87, 5.73, 7.48 V/decade were obtained. The significant increase in the free carrier concentration of the $SnO_2$ films annealed in air led to early turn on. Owing to the large number of free carriers, the channel was formed more easily in these TFTs than in the TFTs fabricated under other conditions. The interface trap concentration and bulk state trap, located at the semiconductor near the interface, concentration are inversely proportional to the SS value, and were lowest for the $SnO_2$ films annealed in air [24]. For annealed $SnO_2$ semiconductors, the trap site resulting from the formation of an OH group or crystal imperfection can be either shallow or deep level states lying in the forbidden bandgap [25]. An increase in the concentration of shallow trap sites (located at the band edge) resulted in degradation of the field-effect mobility and on-current. Similarly, an increase in the concentration of deep level trap sites (located close to the valence band) resulted in degradation of the SS. Therefore, the traps originating from the OH groups or crystal imperfection led to an increase in the concentration of both shallow and deep level trap states. The reduction in the on-current yielded on/off ratios of $10^7$, $10^4$, and $10^4$ for the TFTs annealed in air, vacuum, and $N_2$, respectively.

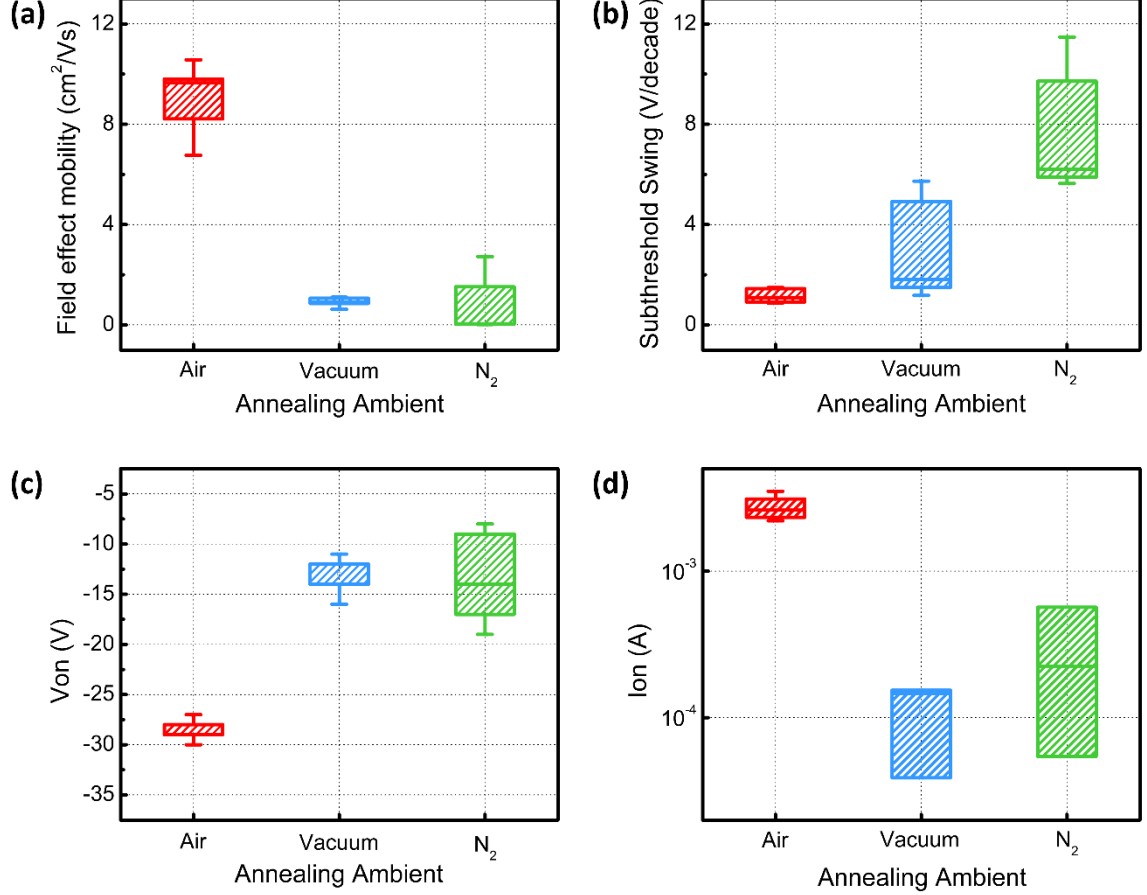

**Figure 5.** Extracted performance parameters of the fabricated $SnO_2$ thin-film transistors: (**a**) field-effect mobility in the saturation regime, (**b**) SS, (**c**) $V_{on}$, and (**d**) $I_{on}$.

## 4. Conclusions

The effect of annealing ambient on $SnO_2$ TFTs fabricated via an ethanol-based sol-gel route was investigated in this study. Annealing conditions have a significant effect on the structural characteristics and chemical composition and, in turn, the device performance. To achieve high-performance $SnO_2$ TFTs, high $Sn^{4+}/Sn^{2+}$ ratios for enhancing the n-type semiconductor properties, and high $O_{vacancy}/OH$ ratios for increasing the free carrier concentration and reducing the trap site concentration are required.

The annealing process in air was helpful in meeting these requirements. The $SnO_2$ films annealed in air had the smallest crystalline-grain size, but exhibited the highest field-effect mobility. Compared with the boundary-scattering minimization achieved by increasing the crystalline size of our system, the free carrier concentration played a more critical role in the realization of high-performance devices. The fabricated $SnO_2$ TFTs delivered a field-effect mobility, SS, and on/off current ratio of 10.87 $cm^2$/Vs, 0.87 V/decade, and $10^7$, respectively.

**Author Contributions:** Conceptualization, W.–Y.L. and J.J.; Experiments and data analysis, H.L. and S.H.; Investigation, H.L., H.H., and K.K.; Writing—original draft preparation, S.H. and H.L.; Writing—Review & Editing, J.H.B., I.M.K., W.–Y.L., and J.J.

**Funding:** This work was supported by the Basic Science Research Program through the National Research Foundation of Korea (NRF) funded by the Ministry of Education (NRF-2016R1D1A3B03930896) and by the National Research Foundation of Korea (NRF) grant funded by the Korean government (MSIT) (2019R1F1A1059788).

**Conflicts of Interest:** The authors declare no conflict of interest.

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
