# Peer review of "Effect of Annealing Ambient on SnO2 Thin Film Transistors Fabricated via An Ethanol-based Sol-gel Route"

_electronics, doi:10.3390/electronics8090955_

Round 1

Reviewer 1 Report

In this paper, the authors investigate effect of annealing ambient on SnO2 TFTs. The SnO2 film annealed in air has the highest electron concentration and subsequently leads to the best TFT performance. The mobility of the TFT is 10.87 cm2/Vs and on/off ratio is 107. The performance is acceptable. However, there are serious issues for the explanation of the TFTs annealed in different ambients. In addition, the explanations are not consistent.  

In Fig. 2(h), can authors explain why the oxygen vacancy content in the SnO2 layer is higher than that of the one annealed in vacuum? Air ambient contains oxygen and can provide oxygen to the SnO2 The result is controversial. In line 142: Compared with that generated under other conditions, more SnO was converted in SnO2 and more oxygen vacancies can be generated more easily after annealing in air. It seems unreasonable for this statement. The authors should give more explanation and give related literature to support this point. The oxygen vacancy content has a great impact on the TFT performance. Why authors do not anneal the SnO film in an oxygen ambient? The SnO2 film annealed in O2 usually has a lower oxygen vacancy content [1]. The result can enrich this manuscript. The authors state that compared with SnO (Sn2+), oxygen vacancies can be more easily generated in SnO2. The Sn4+ content of the SnOx film annealed in vacuum is higher than that of the SnOx film annealed in N2. However, why the oxygen vacancy content of the vacuum annealed SnOx is lower than the N2 annealed SnOx film? The author should explain the difference in transfer characteristics of the SnOx films annealed in N2 and vacuum. Can authors provide Ids-Vgs curves at different Vds? The gate leakage current is strongly recommended to add to the transfer characteristics, especially for the best SnO2 It is strongly recommended that the manuscript should be revised by a native English speaker.

[1]       P. D.M, R. Mannam, M. S. R. Rao, and N. DasGupta, "Effect of annealing ambient on SnO2 thin film transistors," Applied Surface Science, vol. 418, pp. 414-417, 2017/10/01/. 2017, doi: https://doi.org/10.1016/j.apsusc.2016.11.233.

Reviewer 2 Report

This is a well written paper with thorough characterisation and clear conclusions.

Eqn 1 has been corrupted by pdf conversion.

The validity of Eqn 2 should be indicated in the usual manner.

Please explain the negative slope in Fig. 3b.

Conventional silicon FET models (Eqn.2, SS) are assumed although Figs 3c,d do not show clear  exponentional behaviour – more like a power law. Please justify your adoption of the Si models.

You assume SS is solely related to interface states, What about bulk states? Please justify.

Do the terms shallow and deep states refer to the interface or the bulk? Please clarify. There are techniques for extracting these states. Please consider using them so as to improve the discussion.

Round 2

Reviewer 1 Report

The authors do not properly address the issues that the reviewer asked. In addition, the authors delete several figures (Fig. 2(c)-(f)) for avoiding the issues, not for addressing the issues. The manuscript becomes to be seriously flawed. The manuscript should be rejected and rewritten very carefully.

Additional questions for the revised manuscript    

1. The mobilities described in the abstract and the conclusion are not consistent. The authors should carefully check the

2. The interface trap concentration is inversely proportional to the SS value. The paper that the authors provide does not mention that the bulk trap concentration is inversely proportional to the SS value.

3. Why a high Sn4+/Sn2+ ratio leads to a higher carrier concentration?

4. The authors do not explain the difference in the transfer characteristics of the SnOxTFTs annealed in N2 and vacuum.

Round 3

Reviewer 1 Report

This manuscript can be accepted.